# Novel Therapeutic Approaches in Neoplastic Meningitis

**DOI:** 10.3390/cancers15010119

**Published:** 2022-12-25

**Authors:** Atulya Aman Khosla, Shreya Saxena, Ahmad Ozair, Vyshak Alva Venur, David M. Peereboom, Manmeet S. Ahluwalia

**Affiliations:** 1Miami Cancer Institute, Baptist Health South Florida, Miami, FL 33176, USA; 2Faculty of Medicine, King George’s Medical University, Lucknow 226003, India; 3Fred Hutchinson Cancer Center, University of Washington, Seattle, WA 98109, USA; 4University of Washington Cancer Consortium, Seattle, WA 98195, USA; 5Cleveland Clinic Lerner College of Medicine, Cleveland Clinic Main Campus, Cleveland, OH 44195, USA

**Keywords:** neoplastic, meningitis, leptomeningeal, chemotherapy, radiotherapy, intrathecal, immunotherapy

## Abstract

**Simple Summary:**

Neoplastic meningitis (NM) is a frequent complication of cancer and is associated with a poor prognosis. The currently available therapies aim to alleviate symptoms and preserve the quality of life. It comprises a multimodal approach, including surgery, intrathecal and systemic chemotherapy, and radiotherapy. The specific treatment is individualized, based on clinical practice guidelines and expert opinion. There are multiple clinical trials undertaken to evaluate the efficacy of novel therapies, including targeted and immunotherapies. This article presents an updated review of treatment approaches in NM.

**Abstract:**

Central nervous system (CNS) metastasis from systemic cancers can involve the brain parenchyma, leptomeninges, or the dura. Neoplastic meningitis (NM), also known by different terms, including leptomeningeal carcinomatosis and carcinomatous meningitis, occurs due to solid tumors and hematologic malignancies and is associated with a poor prognosis. The current management paradigm entails a multimodal approach focused on palliation with surgery, radiation, and chemotherapy, which may be administered systemically or directly into the cerebrospinal fluid (CSF). This review focuses on novel therapeutic approaches, including targeted and immunotherapeutic agents under investigation, that have shown promise in NM arising from solid tumors.

## 1. Introduction

Neoplastic meningitis (NM), also known as leptomeningeal metastasis or leptomeningeal carcinomatosis, refers to the involvement of the subarachnoid space and leptomeninges- arachnoid and pia mater by primary tumor spread. The incidence of NM ranges from 5–8% in patients with solid tumors to 15% in patients with hematologic malignant spread, and it often accompanies metastases to the brain (BMs) [1]. NM has historically been associated with a dismal prognosis of 2–4 months, and it continues to remain poor, with patients presenting with a wide range of clinical features from simultaneous involvement of multiple locations throughout the neuraxis [1,2]. The diagnosis of NM requires a high index of clinical suspicion and is made by imaging with cerebrospinal fluid (CSF) studies. The management of patients with NM has evolved tremendously over the past decade, improving both the quality of life and survival. This narrative review highlights these advancements in management, focusing on new therapeutic modalities, including targeted and immunotherapies.

## 2. Epidemiology

The incidence of NM is around 5% in patients with metastatic cancer, with most patients being diagnosed late in the disease course [3]. Brain metastases frequently accompany NM in as much as 50–80% of patients with NM [4]. The most common solid tumors resulting in NM include breast cancer, followed by lung cancer, melanoma, gastrointestinal malignancies, and metastases from an unknown primary [5]. Among patients with lung cancer, NM occurs most frequently with epidermal growth factor receptor (EGFR) positive non-small cell lung cancer (NSCLC). Similarly, in breast cancer, tumors harboring the human epidermal growth factor receptor 2 (HER2) are more likely to spread to the leptomeninges [6,7]. Primary parenchymal brain tumors also have the potential to spread through the leptomeninges or via the CSF [8]. Surgical resection and stereotactic radiosurgery inpatients with brain metastases have rarely been associated with consequent NM due to spillage and consequent seeding of malignant cells. A multi-institutional analysis studying radiographic NM subtypes showed a greater risk of neurologic death among the classical NM pattern, as compared to nodular NM [9,10,11]. The risk of leptomeningeal seeding has been greater with the omission of whole-brain radiation therapy (WBRT) and reported more with piecemeal rather than en-bloc resections [12,13]. Finally, with the development of targeted therapies and associated improvements in survival, the incidence of NM with or without BM has increased [14].

## 3. Pathogenesis

The pathophysiology of NM involves a multifactorial process in which tumor cells spread from the primary tumor, traverse the vasculature, and seed at a location where they can enter the CSF. A key process in the pathway is the breakdown of the brain barrier (Figure 1).

Involvement of the leptomeninges, which comprise the arachnoid and pia mater, allows the malignant cells to grow and reach throughout the subarachnoid space via the CSF [16]. While present inside the CSF, the tumor cells are preserved from immune surveillance and attack, which aids in their further proliferation, referred to as the CSF sanctuary phenomenon [17].

There exist various means via which the tumor cells reach the CSF, with hematogenous dissemination being the most common. Other routes include a direct extension from pachymeningeal or dural metastases, infiltration along nerve sheaths, spread from choroid plexus metastases, and rarely, from tumors arising within the meninges itself [18]. Cranial nerve involvement by tumors comes in several shapes or forms, many of which give rise to the cranial neuropathies in neoplastic meningitis and some of which contribute to the causation of neoplastic meningitis as well (Table 1).

The causation of a multitude of symptoms is attributed to various pathophysiological mechanisms, like cerebral edema, due to BBB disruption or direct tumor involvement, leading to cranial nerve and spinal root dysfunction. Invasion of the brain parenchyma can interfere with circulation and cause diffuse cerebral dysfunction. Finally, increased intracranial pressure, due to either mass effect or flow obstruction, leads to hydrocephalus and associated symptoms.

## 4. Clinical Features

NM classically has a multifocal involvement, and despite presenting with a single symptom, a thorough neurologic evaluation reveals further sites of CNS affection. The specific clinical symptoms are attributable to sites of invasion by leptomeningeal disease itself, or to sequela like hydrocephalus. Headache is the most common symptom of NM, the presenting symptom in 30–50% of patients, and can be due to either meningeal irritation or increased intracranial pressure (ICP) [20]. An association with neck stiffness suggests headache due to meningeal irritation, whereas accompanying symptoms of nausea with vomiting and signs including papilledema point towards increased ICP. Various presentations of encephalopathy are also common in NM due to hydrocephalus, seizures, or diffuse cerebral dysfunction. This can present as disorientation, personality changes, confusion, and forgetfulness. Seizures are observed in up to 25% of patients with NM due to parenchymal irritation from invasion, edema, or adjacent leptomeningeal deposits [21]. The occurrence of epileptiform activity is frequently confused with plateau waves, which occur during positional changes and are a marker of increased ICP. These can be associated with positional headache, dizziness, presyncope, or episodes of frank syncope, and their presence should follow a workup for increased ICP [22]. Cerebellar dysfunction is reported in 20% of patients at presentation and can cause midline and lateral cerebellar symptoms [4].

Invasion of cranial nerves in the subarachnoid symptoms leads to a multitude of symptoms due to cranial neuropathies. Diplopia is the most common symptom and can be caused by the involvement of either cranial nerves III, IV, or VI [23]. Trigeminal nerve involvement leads to sensory changes over the face, with a classic presentation of facial numbness known as the “numb chin syndrome.” [24]. Facial and vestibulocochlear nerve involvement leads to weakness of facial muscles and sensorineural hearing loss, respectively, and lower cranial nerve dysfunction causes dysphagia, dysarthria, and hoarseness, due to laryngopharyngeal involvement. Spinal nerve root involvement has also been reported, with resultant radiculopathy and cauda equina syndrome, with lower spinal roots more frequently involved than the cervical roots. Finally, cortical signs are rarely seen and suggest an accompanying parenchymal invasion.

## 5. Diagnosis

A thorough evaluation, including a complete history and physical examination, is pertinent to identify clues of NM’s multifocal involvement. Neuroimaging studies include a gadolinium-enhanced magnetic resonance imaging (MRI) scan of the brain and cervical, thoracic and lumbar spine, or computed tomography (CT) scan with contrast, with the former having greater sensitivity. However, an MRI is less specific than a CSF cytology examination and depicts enhancing foci within the sulci, cisterns, and subarachnoid space in the spine [25]. These findings can be accompanied by ventriculomegaly, and fluid-attenuated inversion recovery (FLAIR) images depict hyperintensity due to increased protein content in the CSF [26].

The lumbar puncture could show an elevated opening pressure, and CSF examination typically reveals elevated protein and low glucose concentrations, lymphocytic pleocytosis. The derangement in all the parameters is uncommon; however, a completely normal CSF examination is rare [27]. Xanthochromia may be seen with hemorrhage from the leptomeningeal deposits, primarily originating from melanoma [28]. The definitive diagnostic finding by identifying malignant cells within the CSF carries a high specificity, but the sensitivity is low due to sampling issues and may necessitate repeat lumbar punctures. Immunohistochemical studies in the CSF yield may assist in diagnosing patients with NM due to unknown primary [29].

The concentration of tumor markers may also carry relevance, as an increase with respect to serum concentration is strongly suggestive of NM. A rise in the concentration of CSF tumor makers more than 2–3% of serum values is unlikely due to simple diffusion or serum contamination unless an increased CSF albumin concentration is also present- which indicates disruption of the BBB [30]. Novel techniques, including identifying circulating tumor cells and cell-free DNA in CSF, carry high sensitivity and specificity and have been increasingly used in the last few years [31]. Evaluation of CSF by flow cytometry accords importance in the assessment of patients with suspected NM due to primary central nervous system lymphoma or other subtypes of non-Hodgkin lymphoma with a propensity for CNS involvement. The CSF examination reveals an elevated protein concentration, lymphocyte-predominant pleocytosis, and decreased glucose levels, with flow cytometry confirming the presence of malignant lymphoid cells [32].

## 6. Management

The management strategies for patients with neoplastic meningitis (NM) aim to improve neurologic function, prolong survival and prevent further neurologic deterioration. A multidisciplinary tumor board is essential to deciding individual treatment strategies, with inputs from a team of neurosurgeons, radiation oncologists, neuro-oncologists, and medical oncologists. The treatment options are broadly categorized into systemic therapies, radiation therapies (RT), and therapies instilled directly into the CSF: intrathecal therapies (IT).

### 6.1. Symptomatic Management

The development of increased intracranial pressure (ICP) is a frequent consequence of NM, owing to the hindrance of CSF outflow via the arachnoid granulations. Medical therapies exploited to curb the increasing ICP consist of acetazolamide to decrease the CSF production from choroid plexus and, occasionally, hyperosmotic agents such as mannitol and hypertonic saline to treat acute symptomatic raised ICP. Mechanical means of ICP reduction may need to be employed in refractory cases, with the ventriculoperitoneal shunt leading to improvements in both acute decompensation and overall survival [33,34]. Theoretical risks of seeding the tumor cells from CSF to the peritoneum exist, but the overall benefit far outweighs the benefits of CSF redirection [35]. Seizures can be managed with various antiepileptic drugs, with levetiracetam being an excellent first-line option, considering its low risk of drug interactions [36]. Similar to patients with BMs, routine prophylaxis with anti-seizure drugs is not advised, as it leads to unnecessary adverse effects associated with these drugs.

### 6.2. Radiotherapy

Radiotherapy in NM serves the role of symptom palliation in managing bulky disease and may improve subsequent penetration of systemic therapies by disturbing the blood–brain barrier (BBB) [37]. It, however, has not demonstrated an improvement in survival. The extent of RT is tailored according to the extent of neuraxis affected, with focal RT preferred to ablate localized bulky metastases while limiting dose-related toxicity. Whole-brain RT (WBRT) is employed more often, especially in the setting of concomitant brain metastases (BMs), but is associated with significant cognitive decline and toxicity, when combined with systemic therapies. The typical regimen of WBRT delivers a radiation dose of 30 Gy in 10 fractions, but an attenuated course is preferred for patients unable to tolerate an increased dose or duration of treatment, commonly with 20 Gy being delivered in 4 Gy fractions [38].

Based on disease distribution, RT can improve function rapidly in symptomatic patients with or without radiographic evidence of disease. NM leading to radiculopathies or cauda equina syndrome causes varying pain levels, weakness, and bowel and bladder involvement and can be effectively treated with prompt lumbosacral RT [39]. Focused skull-base RT can manage different cranial neuropathies, with the routine dose delivered being 30 Gy in 10 fractions. Avoiding radiation to the temporal lobe can avoid subsequent memory deficits and is primarily pursued in patients having received WBRT in the past [40]. Patients with diffuse cerebral involvement present with encephalopathy, and WBRT is utilized to treat NM and concomitant BMs. The subset of patients experiencing symptomatic improvement with radiotherapy is those with a lower radiographic bulk and a shorter duration of symptoms, with those suffering from prolonged symptoms deriving little to no benefit [41].

The use of memantine and hippocampal avoidance techniques has shown promise in reducing the rate of cognitive deterioration in BMs, but its role in NM remains unclear. Finally, the use of an even more extensive approach, craniospinal irradiation (CSI), targets a broader area theoretically but is infrequently employed in clinical practice. Other than its significant radiation exposure to the abdominal organs, CSI destroys a considerable amount of marrow in the vertebral bodies, making future use of immunosuppressive chemotherapy difficult [35]. Irradiation with a proton beam instead may avoid these toxicities. A phase II trial comparing craniospinal proton irradiation with photon-involved-field radiotherapy recently reported favorable outcomes with the use of proton therapy in patients with solid tumor NM. A significant improvement was noted in CNS PFS (7.5 mths vs. 6.6 mths) and OS (9.9 mths vs. 6.0 mths) with no difference in toxicity outcomes with the use of proton irradiation [42].

### 6.3. Intrathecal Chemotherapy

Intrathecal (IT) chemotherapy has the theoretical advantage of delivering anticancer agents directly to the site of solid metastases and allowing sufficient concentration to be administered throughout the CSF in case of diffuse metastases. However, since the diffusion beyond the CSF is limited only to a few millimeters, IT chemotherapy is employed in non-bulky disease and to treat bulky metastases post-radiation. By avoiding the route through BBB, IT chemotherapy allows a lower dose of cytotoxic agents to be delivered, thereby lowering the risks of systemic toxicity. The IT chemotherapy regimens involve three phases: high-dose induction, intermediate-dose consolidation, and low-dose maintenance [43].

The two main routes for the administration of IT chemotherapies include a lumbar puncture (LP) or via a ventricular reservoir (e.g., Ommaya reservoir). The advantages associated with using the Ommaya reservoir include a better distribution throughout the CSF compartment, avoidance of repeated LPs, and thus greater ease of administration. Ideally, A CSF flow study is conducted prior to administering IT chemotherapy to ensure an unobstructed CSF path and optimal drug distribution. The CSF flow studies described by Chamberlain utilized the isotopes Technetium-99 or Indium-111, and if the reports suggested an obstruction, IT chemotherapy was not opted for [44].

The procedural risks associated with both these approaches constitute the main drawback of this treatment route. These include, but are not limited to, CNS infection, cerebral herniation, and CSF leak [45]. The ventricular reservoirs carry additional risks, including catheter misplacement, tip occlusion, and aseptic, chemical, or septic meningitis. Cessation of therapy with the removal of the reservoir may be required to manage these complications. Besides, the drugs administered intrathecally have short half-lives, with concentrations declining to subtherapeutic levels in matter of hours, and complete elimination in 1–2 days [46].

Other serious complications associated with IT chemotherapy include the occurrence of progressive leukoencephalopathy. It typically occurs with intrathecal methotrexate and has a chronic presentation with cognitive symptoms, incontinence, seizures, and gait alterations. The incidence further increases upon the combination with RT, which forms the basis of the recommendation of administering RT and IT methotrexate at least 2–3 weeks apart. Liposomal cytarabine is a sustained-release form of the drug, which requires a biweekly administration, and may achieve more homogenous CSF distribution than the non-liposomal form [47]. However, it did not demonstrate increased survival or response rates compared to IT methotrexate across a clinical trial or retrospective case review [48,49]. In 1998, Glantz and colleagues reported findings from a classic trial comparing liposomal cytarabine versus methotrexate, both injected intrathecally. They found that while median survival was not significantly different, the time elapsed without symptoms or toxicity (TWIST) was significantly higher, being 99 days for cytarabine and 28 days for methotrexate (Table 2).

Besides, conus medullaris syndrome with arachnoiditis occurs with liposomal cytarabine, which may be prevented to some extent by spacing IT and systemic chemotherapy and administering corticosteroids. The occurrence of bone marrow suppression is common with the use of various cytotoxic agents, with folinic acid rescue recommended after the administration of methotrexate [50]. Finally, acute myelopathy, a disastrous complication classically associated with methotrexate use, can present as quadriparesis or locked-in syndrome. A diagnostic spinal MRI study can reveal normal findings or T2-hyperintense lesions in the posterior columns. It may also be caused by the use of thiotepa or cytarabine and can be differentiated from myelopathy due to tumor progression by measurement of myelin basic protein in the CSF, which rises in case of drug-induced myelopathy.

Commonly used drugs included as part of IT chemotherapy are methotrexate, thiotepa, topotecan, cytarabine, and sustained-release liposomal cytarabine, and thus the number of available options is limited compared to those which can be administered systematically. Owing to the dearth of clinical trials conducted to date, it has been difficult to conclude a definite superiority of one agent over another. Moreover, most of the studies have been single-arm studies, using different endpoints and not taking into account the multiple subtypes or histologies of the primary tumor [35].

IT therapy was shown to increase survival up to 7.5 months in a cohort of patients with NM from breast cancer upon being treated with a combinatorial regimen of RT with IT methotrexate, cytarabine, or thiotepa [51]. In another subset of patients with NM due to NSCLC studied by Chamberlain et al., the median survival was 5.0 months, with the same combinatorial regimen utilized [52]. However, the results of the only randomized clinical trial conducted to date comparing the efficacy of IT chemotherapy with systemic chemotherapy revealed no difference in patient survival or neurologic response amongst the groups. Boogerd et al. conducted this study on 35 patients with NM due to breast cancer and noted an increased risk of neurotoxicity with IT chemotherapy. Finally, a retrospective analysis by Bokstein et al. involving 104 patients with NM revealed no benefit and an increased risk of complications with IT chemotherapy when combined with RT and systemic chemotherapy, compared to a regimen excluding IT chemotherapy [53]. Recent results from a phase I/II study of intrathecal trastuzumab in HER-2 positive cancer with NM showed an OS of 8.3 months for patients with any HER-2 positive histology and 10.5 months in HER-2 positive breast cancer [54]. Pharmacokinetic studies depicted a stable CSF concentration of trastuzumab, suggesting promising future studies on the subject [54].

### 6.4. Systemic Therapy

#### 6.4.1. Chemotherapy

The utilization of systemic chemotherapy assumes importance in treating associated systemically active disease while avoiding cognitive decline and procedural complications linked to WBRT and IT therapies, respectively. Systemic agents have shown some activity due to the BBB breakdown in the NM setting, although efficacy depends heavily on tumor subtype and histology. Some commonly used systemic chemotherapy agents utilized are described in Table 3.

Non-targeted agents like methotrexate, a dihydrofolate reductase inhibitor, have historically been used to treat primary central nervous system lymphoma. It has also demonstrated efficacy in treating NM originating from solid tumors like squamous cell carcinoma (SCC) of the head and neck. The use of high-dose intravenous methotrexate demonstrated a significant increase in survival compared to IT methotrexate, when used as a sole treatment for managing NM, with a median survival of 13.8 mths (vs. 2.3 mths) [55]. Thiotepa, a DNA alkylating agent, does cross the BBB, but its use remains limited as an IT agent. Systemic temozolomide, a current standard of care in treating gliomas, has not demonstrated appreciable efficacy in managing NM. The pyrimidine analog, cytarabine, has shown efficacy in treating CNS leukemia, especially in patients with isolated CNS involvement [56].

#### 6.4.2. Targeted Therapies

With an ever-increasing knowledge of driver mutations and molecular targets, the development of targeted therapies has been progressing at a rapid pace. Non-squamous cell lung carcinoma harboring a mutation in epidermal growth factor receptor (EGFR) has been successfully targeted by erlotinib, which has led to extended survival in patients with NM [57]. Another study reported a median survival of 14 months among patients treated with erlotinib for NM associated with NSCLC [58]. Newer EGFR tyrosine kinase inhibitors (TKIs) such as afatinib and osimertinib have demonstrated better CNS penetration and have a critical established role in the management of BMs from EGFR-mutated NSCLC [59,60]. A case reported the efficacy of a combination of afatinib and cetuximab in a patient with NM due to NSCLC, leading to the resolution of NM lesions [61]. The BLOOM study, a recently concluded phase I clinical trial, demonstrated considerable efficacy with osimertinib, a 3rd generation TKI, in NM arising from NSCLC. Among 18 patients with NM, five patients (28%) had a confirmed response, and 14 patients (78%) achieved disease control upon being evaluated by MRI imaging [62].

Regarding NM arising from NSCLC harboring mutations involving the anaplastic lymphoma kinase (ALK) gene, new-generation ALK inhibitor alectinib has shown considerable CSF penetration and activity in NM [63,64]. The efficacy of ceritinib, another 2^nd^ generation ALK inhibitor, has been reported recently in NM caused by ALK-mutated NSCLC as part of the ASCEND-7 trial. In a cohort of 18 patients with NM, the whole-body ORR was 16.7%, with the median PFS and OS being 5.2 and 7.2 months, respectively [65]. Lorlatinib, one of the newer ALK inhibitors, has CNS penetration, as seen by adverse effects, and case reports show initial efficacy [66].

Mutations in human epidermal growth factor receptor 2 (HER2) have been shown to be associated with an increased risk of CNS spread [67]. Trials involving trastuzumab, a HER2 inhibitor, have shown good CNS response in metastatic breast cancer with spread to the brain, but separate results with respect to response in NM are unavailable [68]. Ongoing trials evaluating the efficacy of lapatinib, a dual EGFR, and HER2 inhibitor, in treating NM have completed recruitment, and results are awaited (NCT02650752). In combination with capecitabine, a 5-fluorouracil prodrug, lapatinib has also demonstrated an encouraging CNS response in the LANDSCAPE trial studying efficacy in patients with BMs [69]. Neratinib, a HER2-targeting tyrosine kinase inhibitor combined with capecitabine, demonstrated promising intracranial activity in patients with HER2 overexpressing breast cancer [70]. There are case series of efficacy in leptomeningeal metastases [71]. Preliminary results from a phase II trial evaluating the effectiveness of the combination of tucatinib-trastuzumab-capecitabine in the treatment of NM from HER2+ breast cancer have reported a median OS of 11.9 months in a cohort of 17 patients (NCT03501979) [72]. Retrospective studies of an antibody-drug conjugate, trastuzumab deruxtecan, in patients with HER2 breast cancer and NM showed initial evidence of activity, and more recently, the DEBBRAH trial included patients with both BM and NM, and the published data showed excellent responses in patients with BM while the data for NM is awaited [73].

Mutations in v-Raf murine sarcoma viral oncogene homolog B (BRAF) are prevalent in melanoma, with its subtype BRAFv600E being the most common subtype [74]. Three separate case reports have described the efficacy of BRAFv600E inhibitors, dabrafenib, and vemurafenib, in treating NM arising from melanoma. The clinical trials published so far have focused on BMs from melanoma, but trials evaluating the efficacy of immunotherapies in NM due to melanoma are underway (NCT02939300) (Table 4).

#### 6.4.3. Immunotherapies

A myriad of immunotherapies, ranging from immune checkpoint inhibitors (ICIs) to CAR T cell therapies, have been incorporated into the management of multiple types of tumors. Immune checkpoint blockade, with the use of antibodies targeted against programmed death-1 (PD1), its ligand (PD-L1), or cytotoxic T lymphocyte-associated protein-4 (CTLA-4), leads to disinhibition of T-cells, allowing them to target tumor cells effectively. All three categories of drugs, anti-PD1, anti-PD-L1, and anti-CTLA-4, have demonstrated efficacy against BMs from NSCLC and melanoma [75,76,77].

A limited number of studies have evaluated the impact of ICIs on managing NM. A phase II study evaluated the combination of nivolumab and ipilimumab in 18 patients with NM and reported an OS of 44% at three months. In addition, a complete response was observed in one patient (5.6%), with stable and progressive disease in 7 (38.9%) and 4 (22.2%) patients, respectively [78]. Multiple phase II trials are underway, and the results depicting the efficacy of PD-1 inhibitors pembrolizumab and nivolumab are keenly awaited (NCT02886525, NCT04729348). Meanwhile, studies involving PD-L1 inhibitors durvalumab and avelumab have been started to demonstrate safety and a tolerable dose for treating NM to inspire future studies involving variable combinations of ICIs (NCT03719768, NCT04356222). Hendriks et al. evaluated a cohort of 1288 patients of NSCLC treated with ICIs, among which 19 patients were observed to have NM. A PFS of 2.0 and a median OS of 3.7 months were reported in that cohort. [79]. Finally, Geukes Foppen et al. evaluated a series of 39 patients with NM due to melanoma and reported an abysmal prognosis even after using ipilimumab at 15.8 weeks [80].

Other clinical trials currently ongoing related to NM due to breast cancer are evaluating the efficacy of CAR T cell therapy (HER2 CAR) and a bi-specific antibody (HER2Bi) (NCT03696030, NCT03661424). Preliminary results from the NM cohort of patients receiving abemaciclib, a cyclin-dependent kinase 4/6 (CDK 4/6) inhibitor, in breast cancer revealed a PFS of 5.9 months and an OS of 8.4 months [81]. Regarding the utilization of immunotherapies to treat NM due to melanoma, two separate cohort studies have demonstrated the use of IT interleukin-2 (IL-2) with varying chemotherapy combinations and reported a similar survival of 7.8–7.9 months, respectively. [82,83]. Finally, intrathecal administration of the immunotherapeutic agent nivolumab has also been attempted in a single-arm phase I/Ib trial (NCT03025256) in patients with NM due to melanoma. Preliminary results include a median OS of 42% at six months, 30% at 12 months, and a tolerable side effect profile, with no grade 4 or 5 toxicities [84] (Table 5).

#### 6.4.4. Other Novel Therapies

Clinical trials evaluating ANG1005, or paclitaxel trevatide, are underway among newly diagnosed NM from breast cancer (NCT03613181). It is a taxane derivative designed to cross the BBB and is made from three paclitaxel molecules covalently linked to Angiopep-2 [85]. IT-delivered monoclonal antibodies have been used to deliver selected radiation or therapeutic agents. This approach of targeted radioimmunotherapy has been used in a phase I study evaluating iodine-131 labeled monoclonal antibody 3F8, targeting GD2-positive NM. A sufficient intra-CSF concentration was achieved, without significant toxicity, with three out of thirteen patients with a radiographic response [86]. A phase II study utilizing this agent is currently underway (NCT00445965). The glycoprotein 4Ig-B7H3 is present on various tumors, targeted by iodine-131 labeled 8H9 monoclonal antibody, and is currently being evaluated in phase I clinical trial (NCT00089245) (Table 6).

## 7. Conclusions

Neoplastic meningitis remains a disease process with poor survival outcomes due to the tumor microenvironment, the inherently aggressive nature of the neoplasm, and the restricted delivery of therapeutic drugs due to the blood–brain barrier. Its management remains a challenge due to limited evidence from a small number of clinical trials. This results in various non-standardized treatment regimens, which are personalized according to patient and source tumor characteristics. There is a need for prospective studies focusing on selected histological tumor types and gauging the efficacy of novel therapeutics which have become available within the last few years. The utilization of improved diagnostic biomarkers and an understanding of the molecular differences between the primary site and metastatic disease will lead to the development of targeted therapies. Assessment of these drugs within clinical trials, including patients with NM as sub-groups, will help define better therapeutic management of patients affected by leptomeningeal tumor dissemination.

## Figures and Tables

**Figure 1 cancers-15-00119-f001:**
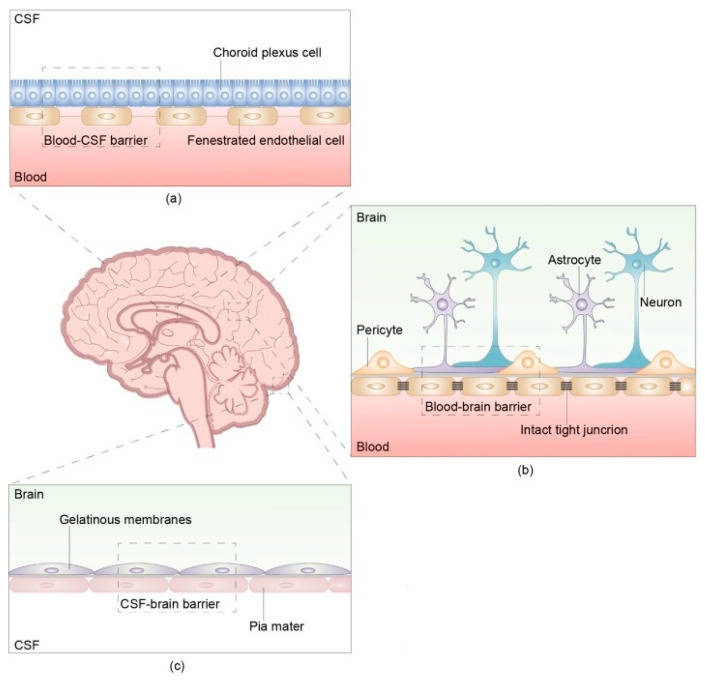
The structure and composition of the brain barrier The brain barrier consists of three parts: the blood- cerebrospinal fluid (CSF) barrier, blood–brain barrier, and CSF-brain barrier. (**a**) The blood-CSF barrier is located between the blood and CSF in the ventricular choroid plexus. (**b**) The blood–brain barrier (BBB) is located between microvascular endothelial cells and the nerve cells of the brain and spinal cord. There are intact tight junctions between capillary endothelial cells that prevent the passage of macromolecules other than water and certain ions. The intact and continuous capillary basement membrane is surrounded by a glial membrane of protruding astrocytes. The BBB, main barrier that protects the CNS, prevents many macromolecules from entering the brain and selectively pumps harmful substances out of the brain. (**c**) The CSF-brain barrier is located between the CSF in the ventricles and subarachnoid space and the nerve cells of the brain and spinal cord. Reproduced with Permission from Wang, Y., Yang, X., Li, N.J., Xue, J.X. Leptomeningeal metastases in non-small cell lung cancer: Diagnosis and treatment. Lung Cancer. 2022;174:1–13. doi:10.1016/j.lungcan.2022.09.013 [15].

**Table 1 cancers-15-00119-t001:** Mechanisms of nerve invasion: several mechanisms of nerve involvement are known and vary from mechanical lesions to different oncological patterns. Abbreviations: NGF, Nerve growth factor; NCAM, Neural Cell Adhesion Molecule. Reproduced with Permission from Grisold W, Grisold A. Cancer around the brain. Neurooncol Pract. 2014; 1(1): 13–21. doi:10.1093/nop/npt002 [19].

Type of Nerve Growth	Subgroups	Remarks
Mechanical causes	Compression	
	Engulfing	
	Pushing and stretching of the nerve by mass lesion	
Invasion	Direct invasion (local infiltration)	
	Perineural	
	Endoneurial	
	Intravascular spread	
Metastasis	Isolated intranerval (rare)	
Perineurial spread	Anterograde	
	Retrograde	
	Particular patterns:	
	Spread via nerve scaffolds	
	Dermatomal spread	
	Anastomotic spread from one nerve region into another	
Tumor invasion—nerve growth	Peripheral nerve sprouting	Observed experimentally
Growth factors	NGF, NCAM, other local factors promoting nerve growth	
Angiosoma vs. common anatomical distribution	Concept of metastatic distribution	The angiosoma concept divides the skull into 13 different angiosomas

**Table 2 cancers-15-00119-t002:** Comparison of key results associated with liposomal cytarabine versus methotrexate, both injected intrathecally. TWIST = time elapsed without symptoms or toxicity. *n* = number of patients. Reproduced with permission from Beauchesne P. Intrathecal chemotherapy for treatment of leptomeningeal dissemination of metastatic tumors. Lancet Oncol. 2010; 11(9): 871–879. doi:10.1016/S1470-2045(10)70034-6 [1].

	Liposomal Cytarabine (*n* = 31)	Methotrexate (*n* = 30)	*p* Value
Response (cytology rendered negative and clinical condition stable or improved)	8	6	0.76
Median duration of response	39 days	26 days	0.31
Time before neurological progression	58 days	30 days	0.0068
Survival directly linked to the meningitis	343 days	98 days	0.074
Median survival	105 days	78 days	0.15
Survival > 6 months	13	5	0.15
Survival > 1 year	5	2	0.43
Grade 3 toxicity	24	20	
Duration of Grade 3 toxicity	18 days	11 days	0.2
TWIST	99 days	28 days	<0.05

**Table 3 cancers-15-00119-t003:** Standard and experimental systemic chemotherapy drugs for treatment of neoplastic meningitis. Reproduced with permission from Beauchesne P. Intrathecal chemotherapy for treatment of leptomeningeal dissemination of metastatic tumors. Lancet Oncol. 2010; 11(9): 871–879. doi:10.1016/S1470-2045(10)70034-6 [1].

	Available for Routine Use	Induction	Consolidation	Maintenance
Methotrexate	Yes	10–15 mg twice weekly	10–15 mg once	10–15 mg once a
		(for 4 weeks)	weekly (for 4 weeks)	month
Thiotepa	Yes	10 mg twice weekly	10 mg once weekly	10 mg once a month
		(for 4 weeks)	(for 4 weeks)	
Cytarabine	Yes	25–100 mg twice	25–100 mg once	25–100 mg once a
		weekly (for 4 weeks)	weekly (for 4 weeks)	month
Liposomal	Yes	50 mg every 2 weeks	50 mg every 4 weeks	
cytarabine		(for 8 weeks)	(for 24 weeks)	
Topotecan	Yes	0·4 mg twice weekly	0·4 mg once per	0·4 mg twice monthly
		(for 6 weeks)	week (for 6 weeks)	for 4 months, then
				monthly thereafter
Mafosfamide	No	20 mg once or twice	20 mg weekly	20 mg every
		weekly until CSF		2–6 weeks
		remission		
Etoposide	Yes	0·5 mg/day for 5 days	0·5 mg/day	0·5 mg/day for 5 days
		every other week	for 5 days every	once a month
		(for 8 weeks)	other week	
			(for 4 weeks)	
Floxuridine	No	1 mg/day continued		
		for as long as possible		
Diaziquone	No	1–2 mg twice weekly		
		for few weeks		
Mercaptopurine	No	10 mg twice weekly for		
		4 weeks		
Busulfan	No	5–17 mg twice weekly		
		for 2 weeks		

**Table 4 cancers-15-00119-t004:** Overview of clinical trials evaluating targeted therapy in neoplastic meningitis. Original table, updated till October 2022. Abbreviations: OS: Overall survival, PFS: Progression-free survival, DCR: Disease control rate, DOR: Duration of response, A/E: Adverse Effects, QoL: Quality of Life.

Study	Targeted Therapy	Primary Site	Estimated Completion	N	Primary Endpoint (s)	Secondary Endpoint (s)
NCT04833205	EGFR-TKI + Nimotuzumab	Lung	April 2023	30	PFS	OS, A/E
NCT04425681	Osimertinib + Bevacizumab	Lung	June 2021	20	PFS, ORR	OS, A/E
NCT04944069	Almonertinib + Bevacizumab	Lung	March 2025	69	OS	PFS, ORR, DCR, DOR
NCT04778800	Almonertinib	Lung	February 2024	60	iPFS	DCR, PFS, OS
NCT02616393	Tesevatinib	Lung	April 2018	36	ORR	PFS, OS, TTP, QoL
NCT05146219	TY-9591	Lung	December 2014	60	ORR	DCR, OS, DOR, PFS
NCT04233021	Osimertinib	Lung	July 2022	113	ORR	OS, PFS, A/E, QoL
NCT03257124	AZD-9291	Lung	December 2021	80	ORR, OS	DCR, OS, DOR, PFS, A/E
NCT03711422	Afatinib	Lung	September 2021	25	PFS, OS, ORR	A/E

**Table 5 cancers-15-00119-t005:** Overview of clinical trials evaluating immunotherapy in neoplastic meningitis. Original table, updated till October 2022. Abbreviations: OS: Overall survival, PFS: Progression-free survival, DLT: Dose Limiting Toxicity, ORR: Objective Response Rate, A/E: Adverse Effects, IT: Intrathecal, CAR-T: Chimeric Antigen Receptor T-cell, N: Number of patients. NSCLC: Non-Small Cell Lung Cancer.

Study	Type of Immunotherapy	Type of Study	Primary Site	N	Therapy	Outcome
NCT02886525	Immune checkpoint inhibitor	Phase II	Multiple	102	Pembrolizumab	ORR, OS, Extracranial ORR
NCT04729348	Immune checkpoint inhibitor	Phase II	Multiple	19	Pembrolizumab + lenvatinib	% alive at 6 mth
NCT03719768	Immune checkpoint inhibitor	Phase I	Multiple	16	Avelumab	Safety, DLT
NCT04356222	Immune checkpoint inhibitor	Phase I	NSCLC	30	Durvalumab	OS, PFS, AEs
NCT03696030	CAR T cells	Phase I	Breast	39	HER-2 CAR-T cells	DLT, AEs
NCT03661424	Immunomodulator	Phase I	Breast	16	Bi-specific antibody (HER2Bi)	AEs: frequency/type/severity/duration
NCT02308020 [81]	Immunomodulator	Phase II	Breast	7 *	Abemaciclib	PFS: 5.9 mth OS: 8.4 mth
Hendriks et al. [79]	Immune checkpoint inhibitor	Retrospective cohort	Lung	19	Pembrolizumab or nivolumab	PFS: 2.0 mth OS: 3.7 mth
Ferguson et al. [82]	Immunomodulator	Retrospective cohort	Melanoma	178	IT IL-2, other combinations with chemo	OS: 7.9 mth
Glitza et al. [83]	Immunomodulator	Retrospective cohort	Melanoma	43	IT IL-2 ± chemoradiotherapy	OS: 7.8 mth
Geukes Foppen et al. [80]	Immune checkpoint inhibitor	Retrospective cohort	Melanoma	10 **	Ipilimumab	OS: 15.8 wks

* 7 patients in cohort with NM, 58 patients analyzed in total. ** 6 patients received ipilimumab, 39 patients analyzed in total.

**Table 6 cancers-15-00119-t006:** Overview of Clinical trials evaluating intrathecal therapies in neoplastic meningitis. Original table, updated till October 2022. Abbreviations: OS: Overall survival, PFS: Progression-free survival, DLT, ORR: Objective Response Rate, RR: Response Rate, N: Number of patients, NSCLC: Non-Small Cell Lung Cancer, SCLC: Small Cell Lung Cancer.

Study	Treatment Arms	Primary Site	N	Outcome
Hitchins [87]	IT methotrexate	SCLC (29%), Breast (25%), 1^o^ brain (9%), NSCLC (7%), lymphoma (7%)	44	ORR: 61%
	IT methotrexate+ IT cytosine arabinoside			ORR: 45%
Grossman [88]	IT methotrexate	Breast (48%), lung (23%), lymphoma (19%)	52	OS: 15.9 weeks
	IT thiotepa			OS: 14.1 weeks
Glantz [49]	IT methotrexate	Breast (36%), NSCLC (10%), 1^o^ brain (23%), melanoma (8%), SCLC (7%)	61	RR: 20%,OS: 78 days
	IT liposomal cytarabine			RR: 26%, OS: 105 days
Glantz [89]	IT cytosine arabinoside	Lymphoma (100%)	28	RR: 15%, OS: 63 days
	IT liposomal cytarabine			RR: 71%, OS: 99 days
Shapiro [90]	IT liposomal cytarabine	Solid tumors (80%), lymphoma (20%)	128	PFS: 34 days
	IT cytosine arabinoside			PFS: 50 days

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
