# Peer review of "Novel Therapeutic Approaches in Neoplastic Meningitis"

_cancers, 2022, doi:10.3390/cancers15010119_

Round 1

Reviewer 1 Report

Authors present a review article on novel therapeutic approaches for neoplastic meningitis, earlier know as meningeomatosis carcinomatosa. The authors need to clearly define what kind of article is this - an experts opinion, letter to the editor, narrative review, scoping review - this also needs to be clearly stated in the Materials and Methods and ideally in the title of the manuscript. Prognosis of patients with neoplastic meningitis is cited from an Lancet article from 2010, with 2-4 months, however we would expect new data if available since our clinical experience is that this might have improved in the course of the last decade. 

Lines 55-57: " Surgical resection in patients with brain me tastases has also been reported to cause more frequent NM due to spillage and consequent seeding of malignant cells" - this is a sentence with a very broad context and should be more thoroughly explained; if it is left without further explanation, it would mislead to conclusion that surgical therapy of brain metastases is something very risky and wrong - and none of it is true. Please explain in which cases is this true and for which subgroup of patients. 

For Pathogenesis there must be more than several sentences, I urge to explain novel molecular mechanisms. 

As for clinical features, the main clinical feature is unfortunately not mentioned - hydrocephalus! This is also the main reason for surgical therapy with ventriculoperitoneal shunt, and this needs to be thoroughly explained. Surgical therapy is also not mentioned, for example Omaya Reservoirs or ventriculoperitoneal or ventriculoatrial shunts. 

Furthermore, I suggest to include subsections on treatment of neoplastic meningitis in different tumor subtypes and the current protocols. 

Author Response

Response to Reviewer 1 Comments

We appreciate the reviewers’ insightful comments. We believe the changes based on these comments and suggestions have improved the manuscript and strengthened our message. We have responded to each of the issues that were raised by the reviewers in a point-by-point manner and are resubmitting the revised manuscript. Specific changes in the text have been called out in the responses detailed below.

  1. Authors present a review article on novel therapeutic approaches for neoplastic meningitis, earlier know as meningiomatosis carcinomatosa. The authors need to clearly define what kind of article is this - an expert’s opinion, letter to the editor, narrative review, scoping review - this also needs to be clearly stated in the Materials and Methods and ideally in the title of the manuscript. Prognosis of patients with neoplastic meningitis is cited from an Lancet article from 2010, with 2-4 months, however we would expect new data if available since our clinical experience is that this might have improved in the course of the last decade. 

Response: We appreciate this feedback from the reviewer. Based on this, we have included the type of review (Narrative) in the introduction itself (line 43) and added an updated citation regarding the prognosis of patients with neoplastic meningitis (Reference 2: Eur J Cancer. 2021)

  1. Lines 55-57: " Surgical resection in patients with brain metastases has also been reported to cause more frequent NM due to spillage and consequent seeding of malignant cells" - this is a sentence with a very broad context and should be more thoroughly explained; if it is left without further explanation, it would mislead to conclusion that surgical therapy of brain metastases is something very risky and wrong - and none of it is true. Please explain in which cases is this true and for which subgroup of patients. 

Response: Thank you for your insightful remark. We agree the remark should have been contextualized. We have incorporated a brief discussion of the case reports where this association between resection of brain metastases and consequent neoplastic meningitis may have likely occurred. In addition, we have included the subgroup of patients in which this occurrence is association is seen.

  1. For Pathogenesis there must be more than several sentences, I urge to explain novel molecular mechanisms. 

Response: We agree with your comment. We have expanded the section on pathogenesis, describing various mechanisms that lead to the diverse clinical features seen in neoplastic meningitis.

  1. As for clinical features, the main clinical feature is unfortunately not mentioned - hydrocephalus! This is also the main reason for surgical therapy with ventriculoperitoneal shunt, and this needs to be thoroughly explained. Surgical therapy is also not mentioned, for example Omaya Reservoirs or ventriculoperitoneal or ventriculoatrial shunts. 

Response: Thank you for the feedback. While the paper had some discussion on hydrocephalus, we have, however, expanded the section on the same. As rightly pointed out, it is indeed the main reason for surgical therapy with ventriculoperitoneal shunt, which has already been mentioned in the section on “Symptomatic Management”, (lines 148-161).

  1. Furthermore, I suggest to include subsections on treatment of neoplastic meningitis in different tumor subtypes and the current protocols. 

Response: We appreciate the feedback and have already incorporated paragraphs describing treatment of neoplastic meningitis in different tumor subtypes within the different treatment subtype headings. For example, under the subheading of targeted therapies, we have included paragraphs on targeted therapies utilized in neoplastic meningitis arising from lung, breast cancers and melanoma, respectively.

Reviewer 2 Report

- Section Epidemiology.  The reference n. 4 is old. Please replace with : Balestrino R, Rudà R, Soffietti R. Brain Metastasis from Unknown Primary Tumour: Moving from Old Retrospective Studies to Clinical Trials on Targeted Agents. Cancers (Basel). 2020 Nov 12;12(11):3350. doi: 10.3390/cancers12113350. PMID: 33198246; PMCID: PMC7697886.

- Section Diagnosis. Line 114: to delete "and positivi serology for malignant cells". Line 121: reference n. 28 is too old : to replace with : Boire A, Brandsma D, Brastianos PK, Le Rhun E, Ahluwalia M, Junck L, Glantz M, Groves MD, Lee EQ, Lin N, Raizer J, Rudà R, Weller M, Van den Bent MJ, Vogelbaum MA, Chang S, Wen PY, Soffietti R. Liquid biopsy in central nervous system metastases: a RANO review and proposals for clinical applications. Neuro Oncol. 2019 May 6;21(5):571-584. doi: 10.1093/neuonc/noz012. PMID: 30668804; PMCID: PMC6502489.

- Section Sympomatic Management. Line 151: to delete "quite common with NM".

- Section Radiotherapy. Line 182-184: to delete "Progressive leukoencephalopathy...".

- Section Intrathecal chemotherapy. There is no mention of the problem of the short half-life of drugs in the CSF.

Author Response

Response to Reviewer 2 Comments

We appreciate the reviewers’ insightful comments. We believe the changes based on these comments and suggestions have improved the manuscript and strengthened our message. We have responded to each of the issues that were raised by the reviewers in a point-by-point manner and are resubmitting the revised manuscript. Specific changes in the text have been called out in the responses detailed below.

  1. Section Epidemiology.  The reference n. 4 is old. Please replace with : Balestrino R, Rudà R, Soffietti R. Brain Metastasis from Unknown Primary Tumour: Moving from Old Retrospective Studies to Clinical Trials on Targeted Agents. Cancers (Basel). 2020 Nov 12;12(11):3350. doi: 10.3390/cancers12113350. PMID: 33198246; PMCID: PMC7697886.

Response: Thank you for your comment. We have replaced reference 4 with the one suggested above.

  1. Section Diagnosis. Line 114: to delete "and positivi serology for malignant cells". Line 121: reference n. 28 is too old : to replace with : Boire A, Brandsma D, Brastianos PK, Le Rhun E, Ahluwalia M, Junck L, Glantz M, Groves MD, Lee EQ, Lin N, Raizer J, Rudà R, Weller M, Van den Bent MJ, Vogelbaum MA, Chang S, Wen PY, Soffietti R. Liquid biopsy in central nervous system metastases: a RANO review and proposals for clinical applications. Neuro Oncol. 2019 May 6;21(5):571-584. doi: 10.1093/neuonc/noz012. PMID: 30668804; PMCID: PMC6502489.

Response: Thank you for the suggestion. We have deleted the suggested text from line 121, and have replaced reference 28 with the updated one.

  1. Section Symptomatic Management. Line 151: to delete "quite common with NM".

Response: Thank you for the feedback. We have deleted the suggested text from line 151.

  1. Section Radiotherapy. Line 182-184: to delete "Progressive leukoencephalopathy...".

Response: Thank you for your comment. We have deleted this line (182-184).

  1. Section Intrathecal chemotherapy. There is no mention of the problem of the short half-life of drugs in the CSF.

Response: We appreciate this feedback from the reviewer. Based on this, we have included the limitation of the short half-lives of the drugs administered intrathecally, and have included a citation as well.

Round 2

Reviewer 1 Report

The authors have sufficiently responded to reviewers remarks.

Reviewer 2 Report

Theauthors hava successfully addressed my concerns